# A new class of hybrid secretion system is employed in *Pseudomonas* amyloid biogenesis

Sarah L. Rouse[1], William J. Hawthorne[1], Jamie-Lee Berry[1], Dror S. Chorev[2], Sandra A. Ionescu[2], Sebastian Lambert[3], Fisentzos Stylianou[1], Wiebke Ewert[1], Uma Mackie[1,4], R. Marc L. Morgan[1], Daniel Otzen [5], Florian-Alexander Herbst[6], Per H. Nielsen [6], Morten Dueholm [6], Hagan Bayley[2], Carol V. Robinson[2], Stephen Hare [1] & Stephen Matthews [1]

Gram-negative bacteria possess specialised biogenesis machineries that facilitate the export of amyloid subunits for construction of a biofilm matrix. The secretion of bacterial functional amyloid requires a bespoke outer-membrane protein channel through which unfolded amyloid substrates are translocated. Here, we combine X-ray crystallography, native mass spectrometry, single-channel electrical recording, molecular simulations and circular dichroism measurements to provide high-resolution structural insight into the functional amyloid transporter from *Pseudomonas*, FapF. FapF forms a trimer of gated β-barrel channels in which opening is regulated by a helical plug connected to an extended coil-coiled platform spanning the bacterial periplasm. Although FapF represents a unique type of secretion system, it shares mechanistic features with a diverse range of peptide translocation systems. Our findings highlight alternative strategies for handling and export of amyloid protein sequences.

---

[1] Department of Life Sciences, Imperial College London, South Kensington Campus, London SW72AZ, UK. [2] Chemistry Research Laboratory, University of Oxford, Oxford OX1 3TA, UK. [3] Duke-NUS Medical School, 8 College Road, Singapore 169857, Singapore. [4] Walthamstow School for Girls, London E17 9RZ, UK. [5] Interdisciplinary Nanoscience Center (iNANO), Centre for Insoluble Protein Structures (inSPIN), Department of Molecular Biology and Genetics, Aarhus University, Aarhus C, Denmark. [6] Center for Microbial Communities, Department of Chemistry and Bioscience, Aalborg University, Aalborg, Denmark. Sarah L. Rouse, William J. Hawthorne and Jamie-Lee Berry contributed equally to this work. Correspondence and requests for materials should be addressed to S.M. (email: s.j.matthews@imperial.ac.uk)

The aggregation of amyloidogenic proteins is typically associated with human disease states such as Parkinson's and Alzheimer's. However, many organisms including bacteria produce extracellular amyloid fibres for beneficial purposes[1]. Bacterial amyloid fibres are a major protein component of biofilms, and play important functional roles in bacterial persistence either in animal hosts or on surfaces[2–6]. Bacterial biofilms are of high societal importance as they are a major cause of recurrent disease by allowing reservoirs of bacteria to persist in a host (animal/human) or the environment and also contribute to drug resistance. Gram-negative bacteria employ sophisticated multi-component secretion systems for amyloid assembly[7]. Amyloidogenic, unstructured subunits cross the periplasm and are secreted through the outer membrane (OM), after which they self-assemble into cross β-strand fibrils upon interaction with an extracellular or membrane-embedded nucleator protein. The aggregation properties of these subunits require strict control until they are translocated to the bacterial extracellular surface[8]. Bacterial amyloid fibres are a major protein component of biofilms, and play important functional roles in bacterial persistence either in animal hosts or on surfaces[2–6]. Two major amyloid systems are known in Gram-negative bacteria: the well-studied curli machinery in *Escherichia coli*[9] and the more recently discovered and genetically distinct Fap system in Pseudomonads[10].

The curli machinery comprises seven proteins, encoded in two operons csgBAC and csgDEFG, of which the mechanistic roles in curli fibre formation have been largely documented[11]. CsgA is the main component of the fibres along with CsgB present at low levels as a nucleator[12–14]. The atomic structure of the CsgG pore-forming component was recently solved and illuminated the molecular basis of transport of CsgA across the membrane[15, 16]. The curli system is distinct from previously described secretion pathways and has been named the Type VIII pathway[7, 17]. The proteins encoded by the *Pseudomonas fap* operon, fapABCDEF, are genetically distinct to curli proteins[10]. However, the amyloid forming component FapC contains similar amyloid repeat sequence rich in asparagine and glutamine residues (GN-X-G-N-X₂- AG-X₂-NQQ-X-N), and inhibitors of curli formation have also been shown to inhibit Fap fibril formation[18]. FapC is the main fibre-forming component, with FapB and FapE present in the fibres as minor subunits[19, 20]. FapA and FapD appear to be accessory proteins with regulatory roles. FapA has been implicated as a chaperone of Fap amyloid secretion, as its absence alters the composition of Fap fibres, which become primarily composed of the minor component FapB[19]. FapF is the membrane protein pore component through which the FapB, FapC and FapE substrates are secreted[20].

Unlike the well-characterised curli system, no structural information is available for any component of the Fap system. Therefore, to provide insight into mechanistic basis of Fap secretion, we solved the crystal structure of the transmembrane domain (TD) of the transporter FapF (FapF₈₃₋₄₀₆), termed FapF_β from hereon. We then used a combination of biophysical techniques including native mass spectrometry (MS), single-channel current recordings and circular dichroism in order to elucidate the role of the periplasmic N-terminal domain of full-length FapF (FapF₁₋₄₀₆) and derive a structural model for the translocation complex. Based on this new insight we designed several mutants to probe the functional determinants of FapC secretion. Combining our structural data with computational and functional analyses we shed new light on the mechanism by which FapC is secreted across the OM. Furthermore, our findings contribute to a growing understanding of how bacteria can safely handle amyloidogenic polypeptides and

provides the inspiration for new approaches in the control of bacterial biofilms.

## Results

**Architecture of the trimeric FapF transporter.** The full-length FapF species (numbered 1–406 for the mature protein from uniprot accession C4IN73) did not afford highly diffracting crystals, despite exhaustive attempts to optimise protein micro-crystals. Therefore, a combination of limited proteolysis and secondary structure prediction was used to produce a series of constructs that generated improved crystals[21]. Removing the N-terminal domain comprising a 39-residue helical region followed by a 42-residue disordered linker and adding an OmpA signal sequence and a hexahistidine tag to allow purification directly from the *E. coli* outer membrane (OM) led to larger, well-diffracting and reproducible crystals for FapF_β (Fig. 1a). Incorporating a L273M mutation into this construct yielded selenomethionine-substituted crystals that produced sufficient anomalous signal for phase determination. The structure was solved by single-wavelength anomalous dispersion (SAD) to 2.5 Å resolution with an Rfree of 26% (Supplementary Table 1 and Supplementary Fig. 1).

The TD of FapF comprises a 12-stranded plugged β-barrel (FapF_β), which forms a tightly packed trimer within the crystal (Fig. 1b and Supplementary Fig. 1). The trimer interface is extensive, with a prominent phenylalanine ladder (F325, F335 and F367) making hydrophobic contacts between protomers around the C3 axis. At the extracellular rim of the FapF trimer is the presence of two conserved 'PTG' motifs at the base of loop β5–β6 (residues 233–235 and 259–261, which introduces a distinct plait in the polypeptide chain (Fig. 1b).

A notable feature of the structure is the presence of a 13-residue helical plug between residues A89 and G101 that is inserted into the lumen of the pore from the periplasmic side with the C-terminal end pointing in and the N terminus out. Strikingly, the helix is immediately followed by a turn with a double phenylalanine motif (¹⁰²FF¹⁰³) before the N terminus loops back into the periplasmic space (Fig. 1c). This hairpin structure constricts the pore and represents a closed conformation, which would not likely allow passage of molecules larger than water through the pore. An evolutionarily conserved salt-bridge interaction between E98 and K401 pins the plug to the internal wall of the barrel (Fig. 1c). The plug helix is also flanked on one side by a specific electrostatic interaction between E111 and R157 linking neighbouring strands within the barrel lumen (Fig. 1c). It is conceivable that a rearrangement of the three salt-bridge partners could facilitate movement of the helical plug and opening of the pore. Molecular simulations of the FapF_β trimer in lipid bilayers (Fig. 1d) confirm that the plug is relatively rigid on a 100 ns timescale and no ion flux occurs (Supplementary Fig. 2). A substantial conformational change must therefore occur to open the channel before a polypeptide can be secreted, which is in stark contrast to secretion through the open CsgG barrel in which a single 9 Å ungated aperture is present[16, 22] (Supplementary Fig. 3).

Our crystal structure lacks the N-terminal region of FapF comprising the first 81 residues which would be predicted to lie within the periplasm. This region harbours a coiled-coil motif with a predicted preference for a parallel trimer[23, 24], consistent with the trimeric architecture of the FapF transmembrane domain observed in our crystal structure. To explore the multimeric nature of FapF further we performed native mass spectrometry of the truncated FapF_β and full-length FapF (Fig. 2a). FapF_β (FapF₈₃₋₄₀₆) used for crystallography was observed to exist in a monomer/dimer/trimer equilibrium at ~3 : 2 : 1 ratio in

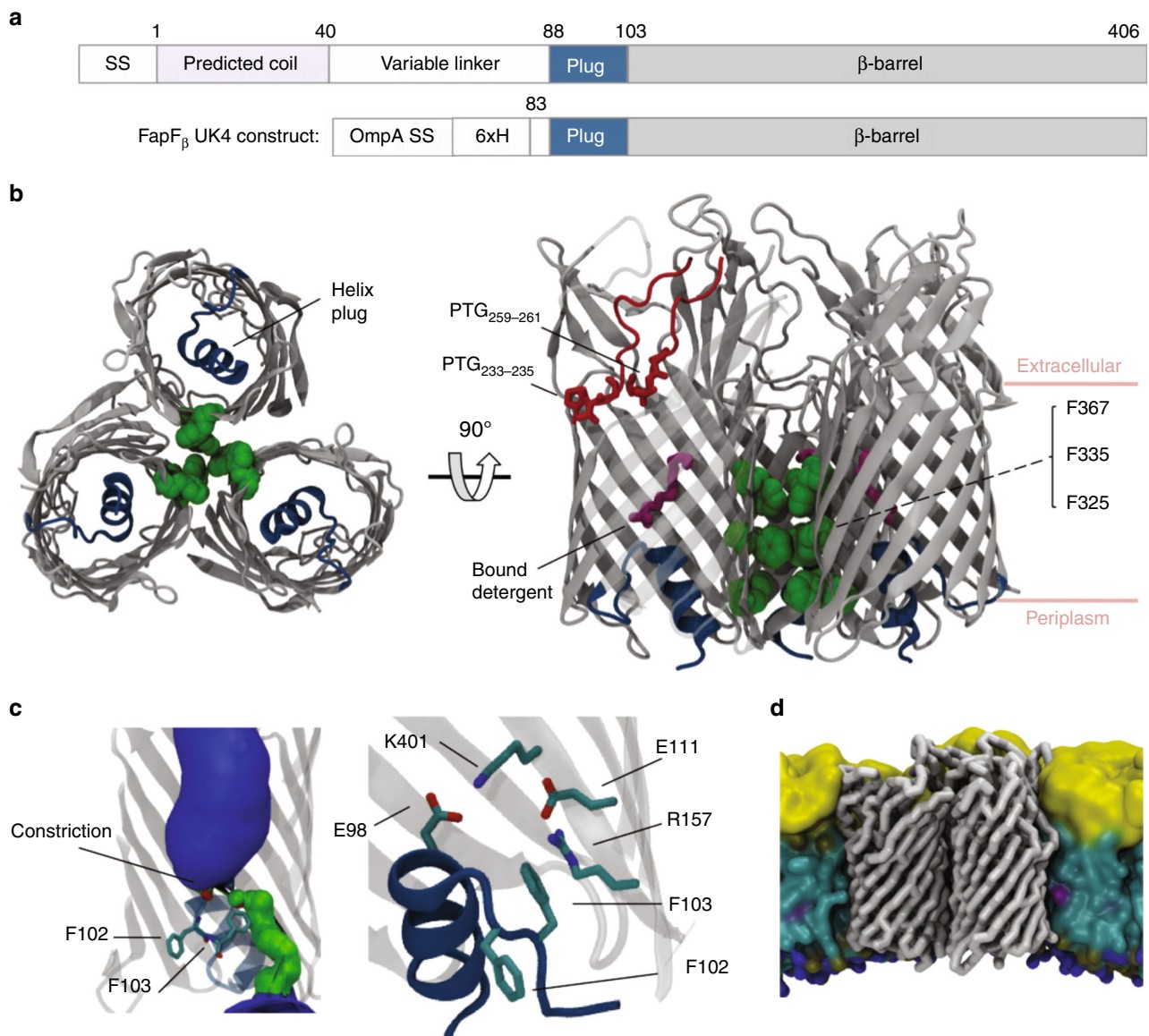

**Fig.1** Overall architecture of the FapF$_\beta$ transporter domain. **a** Sequence of FapF showing approximate domain boundaries and the construct used for structure solution which consisted of the OmpA signal sequence (SS) and an N-terminal His-Tag (6xH) followed by residues K83–F406 of FapF. **b** The crystal structure of FapF$_\beta$. View from the periplasm through the plane of the membrane and perpendicular to the plane. The lines of three closely packed phenylalanine residues are shown in *green space-filling representation*, and the small helix plug is shown in *blue cartoon representation*. Bound detergent molecules in the pore are shown as *magenta ball-and-stick representation*. The 'PTG' motifs (residues P233–G235 and P259–G261) and loop between are shown in *red*. **c** Closer view of the constriction site at the top of the helical plug. Conserved residues are shown in *stick representation*. The pore profile is shown as a surface where *blue* corresponds to bulk water (radius > 2.3 Å), *green* is single-file water (1.15–2.3 Å) and *red* is closed (< 1.15 Å). **d** The equilibrated position of FapF domain within a coarse-grained asymmetric LPS-containing lipid bilayer

C8E4 detergent solution (Fig. 2a). In contrast, for the full-length species (FapF$_{1-406}$), the mass spectrum showed that the most abundant species was trimeric with a small amount of monomer and no dimeric species, suggesting that native FapF is indeed a trimer.

We then reconstituted both FapF constructs into planar DPhPC bilayers. Full-length FapF inserted stably into the bilayer and exhibited a pronounced current–voltage asymmetry, giving a current of $74 \pm 5$ pA at $+100$ mV and $-106 \pm 4$ pA ($n = 21$) in buffer containing 0.1 mM KCl (Fig. 2b). The conductance of the full-length FapF trimer recorded in 1 M KCl was $4.95 \pm 0.03$ nS (Supplementary Fig. 4). All recorded channels exhibited a higher conductance at negative voltages, implying unidirectional insertion. FapF$_\beta$, which contains a plugged barrel in the crystal

structure, required the presence of 4 M urea in the recording chamber in order to obtain insertion of stable channels in an open state for conductance measurements. Urea-induced channel opening in planar lipid bilayer current recordings has been previously reported for plugged OM TonB-dependent transporters[25], and it is assumed that the urea unfolds the plug, opening an ion-conducting pathway through the barrel. After the urea in the recording chamber was exchanged with fresh buffer, the conductance measured for FapF$_\beta$ did not significantly differ from the full-length trimer: $73 \pm 2$ pA at $+100$ mV and $-103 \pm 4$ pA at $-100$ mV in 0.1 M KCl ($n = 10$) (Fig. 2b). These observations suggest the plug is labile in the full-length FapF, which exists predominantly in an open state. The fact that we were unable to observe channel activity for

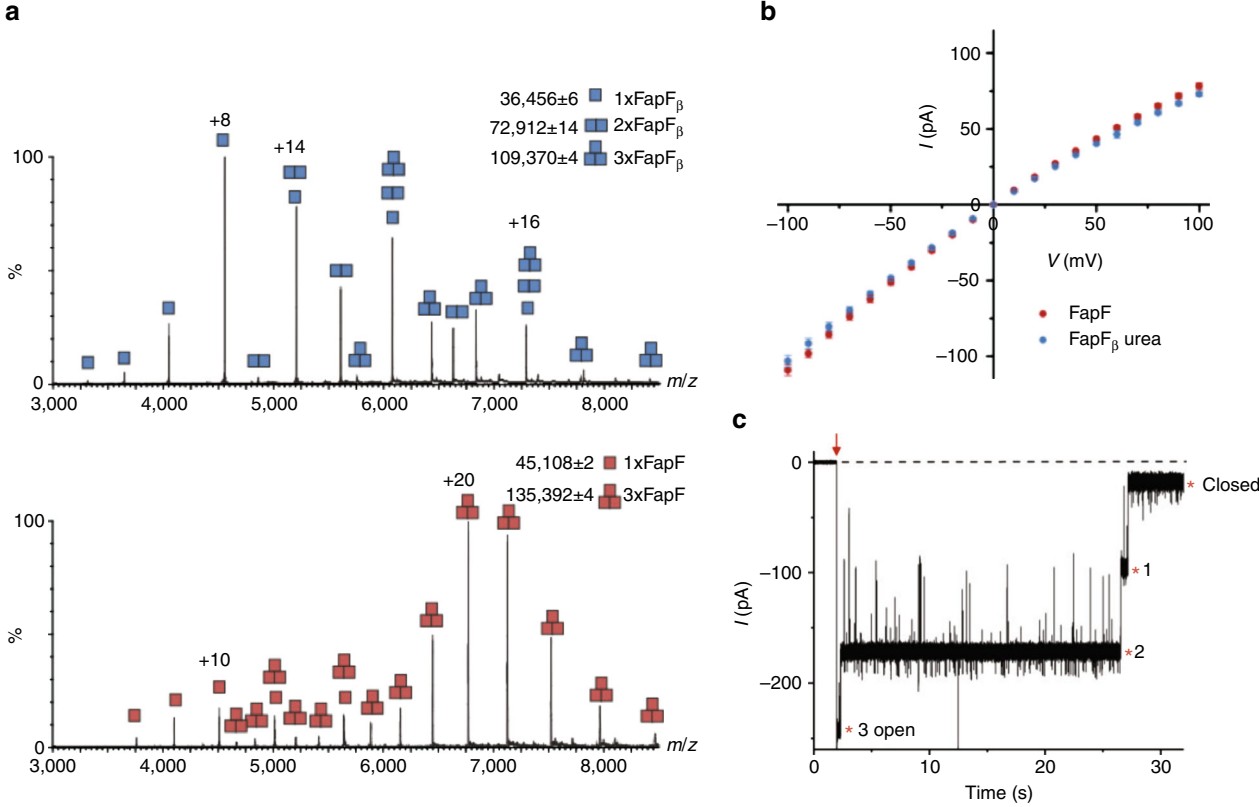

**Fig. 2** Native mass spectrometry analysis and single-channel electrophysiology of FapF. **a** The oligomeric state of FapF$_\beta$ (data shown in *blue*) and full-length FapF (data shown in *red*). **b** *I*–*V* curves for the full-length FapF (*red*, $n = 11$) and FapF$_\beta$ (*blue*, $n = 6$) reconstituted into planar DPhPC bilayers. The curves display mean unitary conductance values ($\pm$s.d.) from $-100$ mV to $+100$ mV in 0.1 M KCl, 20 mM potassium phosphate buffer, pH 7.0. FapF$_\beta$ required the addition of 4 M urea to the recording chamber, which presumably unfolds the plug to enable insertion of an open channel. The urea was exchanged with fresh buffer before measuring conductance. **c** Single-channel gating trace of full-length FapF. At an applied potential of $-250$ mV (*red arrow*), full-length FapF gated in three equal steps (marked with an *asterisk*) which is indicative of a trimer. The *dashed line* represents the baseline at 0 mV

FapF$_\beta$ in the absence of urea indicates a highly stable plugged conformation, and may suggest that the crystal structure corresponds to a final step in secretion in which the channels are stably closed. The removal of urea did not lead to a decrease in FapF$_\beta$ conductance, which implies that the helical plug does not refold into the channel lumen. Irreversible plug denaturation was previously observed for the OM vitamin B12 receptor BtuB[25]. At applied potentials exceeding $\pm 200$ mV, both the full-length and FapF$_\beta$ constructs exhibited voltage-induced closure in three steps (Fig. 2c), which has been previously observed in trimeric OM porins[26]. This further suggests that native FapF exists in the bilayer as a trimer. In further support of the FapF trimer being the native form, all attempts to generate a monomer by mutation of the trimer interface in order to generate a monomer, notably by disruption of the phenylalanine ladder (F325, F335 and F367) (Supplementary Fig. 5), prevented FapF$_\beta$ from being isolated from the OM.

To investigate the structure of the putative coiled-coil region we performed circular dichroism (CD) spectroscopy. CD spectra were collected of both full-length FapF and a synthetic peptide D3–Q40, which corresponds to the predicted coiled-coil region (Fig. 3a). These data indicated a high helical content (64%) for the coiled-coil peptide alone (Fig. 3b) that was stable up to 80 °C. Furthermore, full-length FapF showed an increased helical content than would be expected for the FapF$_\beta$ domain alone (Supplementary Fig. 6), consistent with the formation of the helical coiled-coil region. The oligomerisation

state of the isolated coiled-coil region peptide was determined by size exclusion chromatography using a Superdex 200 HR 10/30 column coupled to multiple laser light scattering (SEC-MALS). The molecular mass of the D3–Q40 peptide (theoretical MW=4.5 kDa) in solution was determined to be 13.5 kDa$\pm$5% (Fig. 3c) which, together with the CD data, confirm a coiled-coil trimeric state of the N-terminal domain of FapF. The high stability of the N-terminal coiled-coil region would likely trimerise and localise within the periplasm, as predicted from our crystal structure of FapF$_\beta$. To provide further evidence we first immunoblotted for the N-terminal His-tag before and after cell lysis (Supplementary Fig. 7a). Only after lysis was the tag accessible to antibody detection. We also created an N-terminal FapF fusion to the globular β-lactamase and probed for ampicillin sensitivity, which has use to infer correct folding of fusion and periplasmic localisation[27, 28]. Ampicillin resistance was observed for *E. coli* cells, which expressed the FapF fusion, but not in those that produced the wild-type protein only (Supplementary Fig. 7b). The most likely explanation is that the N-terminal β-lactamase domain of the FapF fusion successfully folds and is retained with the periplasm.

To estimate the overall dimensions of full-length FapF we generated a structural model for the trimeric transporter. The N-terminal coiled-coil region (D3–Q40) was first modelled as an archetypal parallel trimer. The lowest energy model was chosen and shown to be stable by independent simulations (Supplementary Fig. 8). We then combined the N-terminal coiled

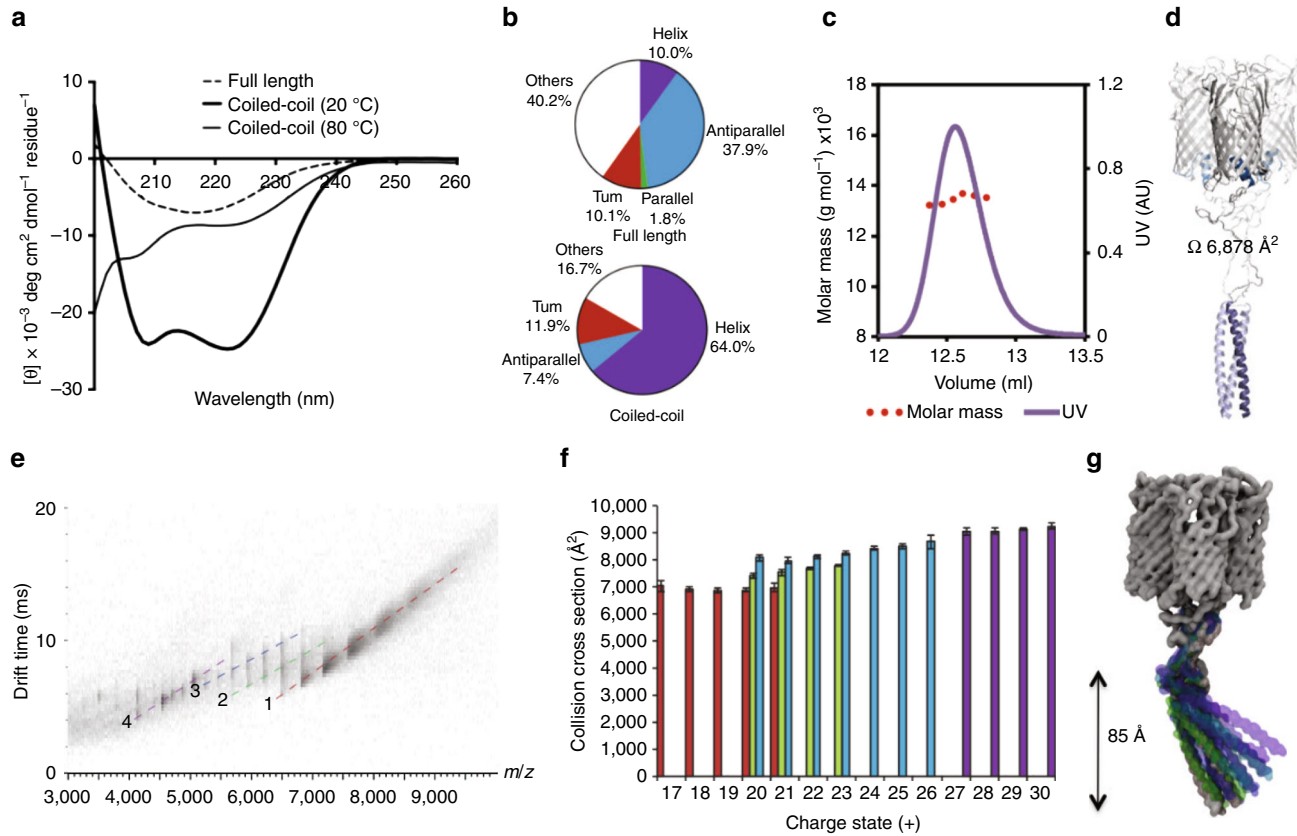

**Fig. 3** Biophysical characterisation of the FapF N-terminal coiled coil. **a** Far-UVCD spectra for the full-length FapF (*dashed line*) and the coiled-coil peptide (D3–Q40) at 20 °C (*thick solid line*) and 80 °C (*thin solid line*). **b** Secondary structure composition for full-length FapF and the coiled-coil peptide (20 °C) as estimated by the BeStSel method. **c** SEC-MALS profile of the coiled-coil indicating a monodisperse sample corresponding to the trimer. **d** Atomistic model of full-length FapF coloured corresponding to the structural boundaries shown in Fig. 1a. The theoretical CCS of this model is shown. **e** Ion-mobility measurements for purified full-length FapF. These indicate a population of four main substate populations. **f** Collision cross-sections of the four different FapF trimer populations. *Error bars* are derived from an average of three different wave heights. **g** Motion of the N-terminal domain of full-length FapF observed during coarse-grained simulations. The trajectory was aligned to the barrel and the position of the N-terminal domain at 10 ns intervals are shown (*grey*=0 ns, *green*=10 ns, *blue*=20 ns, *purple*=30 ns)

coil trimer with our trimeric structure of the FapF$_\beta$ β-barrel and plug (K83–F406) by introducing a disordered linker (P41–L82) to produce a complete model of full-length FapF (Supplementary Fig. 9). Our model predicts a cross-section of 6,878 Å$^2$ (Fig. 3d). This value was compared with data from collision cross-section measurements using ion mobility mass spectrometry. The ion mobility data revealed the presence of four distinct conformations (Fig. 3e). The predominant form has a cross-section of 6,863–7,035 Å$^2$ in excellent agreement with our model (Fig. 3b, c and Supplementary Table 2). The remaining three conformational states, corresponding to larger collision cross-sections, indicated that the full-length species can adopt more extended forms (Fig. 3f). This could indicate that the flexible linker region adopts a more extended conformation than suggested by our model, in which the coiled-coil domain is further from the barrel. Coarse-grained molecular dynamics simulations demonstrate that N-terminal coiled-coil domain together with the linker is highly elongated and mobile relative to the transmembrane barrel (Fig. 3g), and could potentially span the entire depth of the periplasm[29], interacting with and/or anchoring beneath the peptidoglycan layer.

Lipid interactions are known to play a major role in protein stabilisation within the membrane, but structural information is sparse[30]. We observed electron density in the crystal structures that can be attributed to carbon chain density; however, only the bound detergent molecules shown in Fig. 1c were conserved

between all three protomers (Supplementary Fig. 10). We performed molecular simulations of FapF$_\beta$ in a simple palmitoyloleoyl phosphatidylglycerol/palmitoyloleoyl phosphatidylethanolamine (POPG/POPE) membrane to further probe the influence of lipid binding on the behaviour of FapF$_\beta$. The presence of POPG and POPE was found to disrupt the hydrogen bonding between strands at the plait near the PTG motifs. This leads to a larger gap between strands β5 and β6 (Supplementary Fig. 11). Furthermore, simulations of FapF in an asymmetric lipopolysaccharide (LPS)-containing membrane revealed putative LPS lipid-binding sites located on the extracellular loops of FapF (Fig. 1d and Supplementary Fig. 12). The presence of LPS-binding sites has recently been shown to be vital for biogenesis of Gram-negative OM proteins[31]. The conserved PTGs are interesting structural motifs as one of these induces a marked twist in the major extracellular loop β5–β6. It is conceivable that this forms part of an exit gate or regulatory role for secretion.

**Coiled-coil and plug domains are critical for FapC secretion.** The large N-terminal coiled-coil domain assists full-length FapF in forming highly stable trimers. Single-channel current recordings suggest an open conformation of the full-length trimer compared to the FapF$_\beta$ trimer, suggesting that all three plugs are readily released in the full-length trimer. Further, the associated

**a**

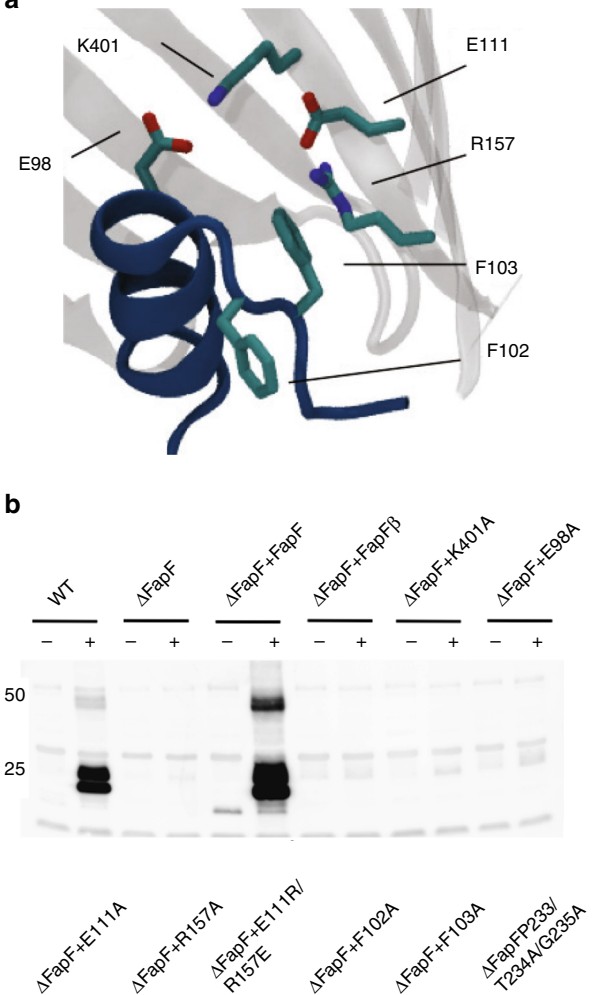

**b**

Fig. 4 Heterologous FapC amyloid production in *E. coli*. **a** Position of residues assessed in this work within the crystal structure. **b** Shown are secretion assays for the Δ*fapF* operon complemented with full-length, wild-type FapF, Δ*fapF* operon complemented with FapF$_\beta$ and various point mutations of FapF within the operon itself. These were treated with either fomic acid (+) or water (−). If FapC is secreted to form amyloid, it is detected by αFapC as a band at ~23 kDa after formic acid treatment. A second band is also produced of slight lower molecular weight which likely reflects a degradation product. Approximate MW markers of 50 and 25 kDa are shown as indicated. Full, uncropped gels are shown in Supplementary Fig. 13

**Table 1 FapC secretion activity of FapF mutants described in Fig. 4**

| FapF mutant | FapC secretion |
| --- | --- |
| E98A | − |
| F102A | − |
| F103A | + |
| E111A | + |
| R157A | − |
| E111R/R157E | + |
| K401A | − |

blot analysis of whole cells with or without pretreatment with formic acid, which depolymerises the extracellular amyloid for sodium dodecyl sulphate–polyacrylamide gel electrophoresis (SDS–PAGE) separation. The secretion assay was performed with the wild-type *fap* operon and a Δ*fapF* derivative that was complemented with either full-length FapF or our truncated crystal construct, FapF$_\beta$, on a separate plasmid. In the absence of FapF, only intracellular FapC could be detected. While the Δ*fapF* mutant operon could be successfully complemented with full-length FapF, the N-terminal truncation FapF$_\beta$ could not support secretion, although FapF$_\beta$ was able to stably insert into the OM of *E. coli* (Fig. 4). This highlights a role for the N-terminal periplasmic regions in regulating secretion through FapF, likely by the controlling plug conformation. It is noteworthy that in secretion assays, two bands are visible for formic acid-treated FapC amyloid (Fig. 4). This has not been observed previously in immunoblots of *Pseudomonas* FapC secretion[20] and therefore likely reflects a stable degradation product due to the presence of *E. coli* proteases. Furthermore, only small amounts of intracellular FapC can be detected mutant expressing the Δ*fapF* operon, suggesting that it is degrading by periplasmic proteases in *E. coli*. This is also observed for FapF mutants that do not secrete amyloid, suggesting that FapC monomer degradation is rapid. Some secretion-competent mutants display increased intercellular FapC, which indicates that amyloid secretion is perhaps less efficient and allow a temporary build-up of FapC (Fig. 4).

We next set out to shed light on the structural and functional importance of the helical plug interactions by generating FapF mutants for complementation of the Δ*fapF* strain that we anticipated would interfere with the packing and/or the dynamics of the plug domain within the pore. Immunoblotting for the His-tag of the mutant proteins confirmed their expression (Supplementary Fig. 13). Mutation of the salt-bridge interaction between E98 and K401, which pins the plug to the barrel wall, abolishes FapC secretion (Fig. 4 and Table 1). Two further conserved charged residues that engage in an interstrand salt bridge within the barrel lumen (E111 and R157) were also shown to contribute to FapC secretion: while the E111A mutant displays secretion activity, the R157A is secretion deficient. Reversing the charge in this salt bridge, with the double mutant E111R/R157E, restores secretion to normal levels, indicating the presence of an arginine residue in this area of the plug is vital for amyloid production (Fig. 4 and Table 1).

We also mutated the conserved double-phenylalanine residues ($^{102}$FF$^{103}$) that stabilise the position of the plug within the barrel (Fig. 4 and Table 1). Interestingly, F103A FapF was fully functional, secreting FapC in an amyloid-competent form, whereas mutating the neighbouring F102 to alanine could not rescue amyloid secretion in the ΔFapF background. The critical F102 is packed within residue side chains in the barrel wall, anchoring the helix within the pore and defining the conformation of the hairpin within the barrel. In contrast, the dispensable F103 makes cation-π contacts with the residue R157. Steered

periplasmic domain may regulate the interaction of the helix plugs with the interior of the β-barrels.

It has previously been shown that the whole *fap* operon can be expressed in *E. coli*, thereby enabling the bacteria to form Fap-dependent biofilms[10]. We adapted this system to test mutant operons for secretion. Secreted FapC is rapidly transformed into the amyloid state outside the cell, whereas intracellular FapC remains in a monomeric state. Secreted FapC can therefore be detected using FapC-specific antibodies and western

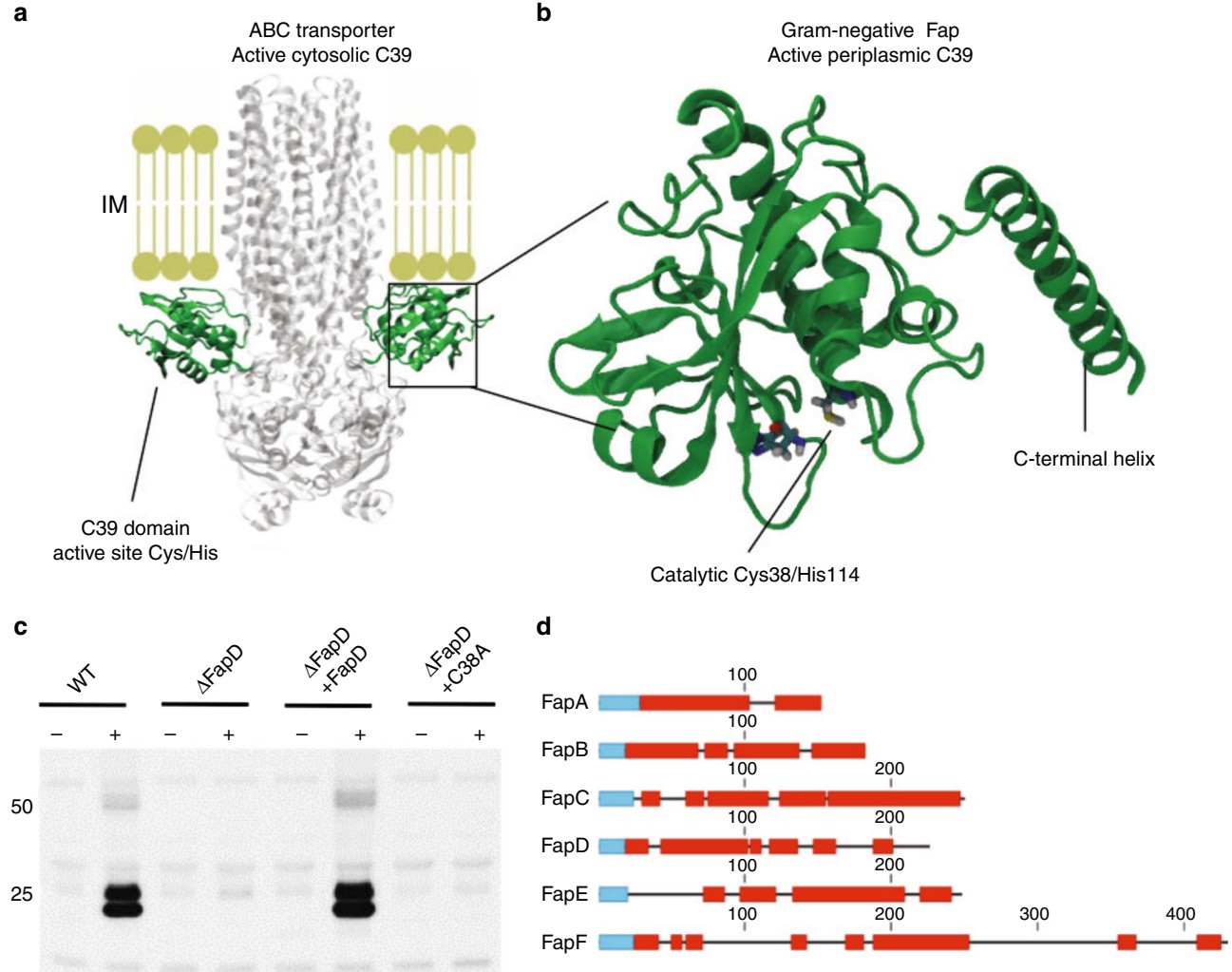

**Fig. 5** FapD is a periplasmic C39 protease required for FapC secretion. **a** Comparison of C39 domains in peptidase-containing ATP-binding cassette transporters. **b** FapD homology model generated by i-Tasser from PDB id 4ry2 (*green*). The active site His and Cys are conserved and shown in *stick representation*. **c** A C38A mutant was unable to secrete FapC fibres demonstrating FapD functions as an active peptidase. Coexpression of the FapD knockout operon with a plasmid encoding wild-type FapD with an OmpA signal sequence restores FapC secretion and confirms FapD is active in the periplasm. The 25 and 50 kDA MW markers are shown. **d** Mapping of peptides observed in whole-cell lysates of *Pseudomonas* sp. UK4 expressing the complete *fap* operon to the Fap protein sequences. Regions with matching peptides are shown as *red boxes* and the signal peptides, which are absent in the mature proteins, are shown as *blue boxes*. For additional details, see the Supplementary Data 2. Peptides are observed for FapA–D throughout the sequence indicating no processing. No peptides were observed in the mature N-terminal of FapE, although sequence analysis indicate the possibility for several theoretical peptides of proper size. This indicates a proteolytic processing of FapE. For FapF the disordered linker region residue 60–100 is potentially processed whilst the putative coiled-coil region remains intact.

molecular dynamics simulations of FapF$_\beta$ were performed to 'pull' the helix plug from the pore (Supplementary Fig. 14). These indicated that the F102 is a key anchoring point for the helix plug domain within the barrel lumen. Furthermore, unbiased molecular simulations of the F102A/F103A double mutant showed that the whole plug is more mobile, and in particular a helix begins to move out from the barrel in one of the three subunits within the simulation timescale of 100 ns (Supplementary Fig. 15), leading to an opening of the channel. Both mutations were introduced into FapF$_\beta$ and purified as for the wild type. Suitable yields for crystallisation were not obtained for FapF F102A; however, a crystal structure was determined for FapF F103A mutant in which the same overall plug placement was observed within the pore as seen in the wild type (Supplementary Fig. 16).

Intriguingly, although R157A crystallises under the same condition as the wild type, we found the symmetry of the trimer was disrupted. The structure was solved at 3.2 Å. Despite the more limited resolution, it is clear that the helix plug remains in position. However, in a single monomer of each trimer there is a conformational rearrangement of side chains due to the void generated from removal of the large R157 side chain. The side chain of F103 is observed to shift (Supplementary Fig. 17), and extra density is observed that may be attributed to a detergent molecule binding in this region. While detergent binding is not necessarily relevant for the in vivo activity of FapF, these data together with simulations imply that each pore of the trimer can function independently.

**FapD is a cysteine protease necessary for Fap secretion.** Sequence analysis of FapD revealed sequence homology with the C39 family of cysteine peptidases, which are usually present in Type 1 secretion systems (T1SSs) for the transport and

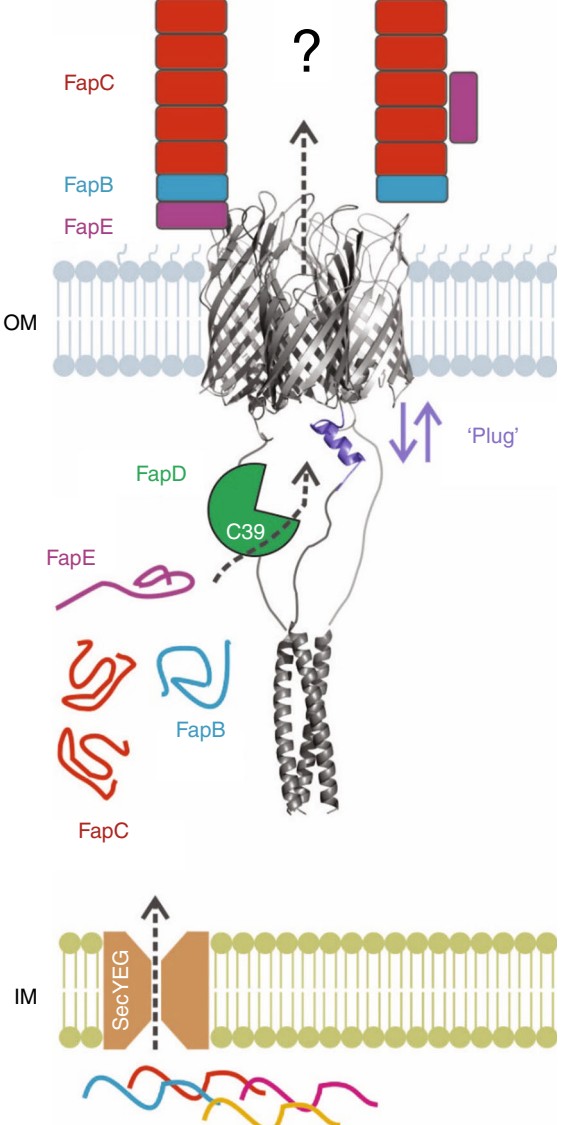

**Fig. 6** Model of functional amyloid secretion in *Pseudomonas*. Fap components are secreted through the inner membrane (IM) via the Sec pathway. FapB (*blue*), FapC (*red*) and FapE (*magenta*) remain unfolded in the periplasm. FapF (*grey*) forms a stable trimer within the outer membrane (OM). The N-terminal coiled-coil of FapF resides in the periplasm. Conformational change of one or more FapF 'plug' domains allows substrate secretion through the OM. For simplicity we show a single plug (*purple*) motion here. FapD (*green*) proteolytic activity essential for secretion. Two suggested models for the architecture of FapE, FapB and FapC in fibres on the extracellular surface of the OM

maturation of bacteriocin precursors[32] (Fig. 5a, b). These peptidase domains associate with the cytosolic faces of ATP-binding cassette of the ABC transporters[33] and usually cleave substrate polypeptides at a GG motif (and sometimes GA) before they are translocated across the plasma membrane[32]. FapD contains a Sec signal peptide and therefore resides in the periplasm rather than the cytosol, accordingly, it likely cleaves a component of the Fap system during secretion. To assess whether FapD is essential for FapC secretion we tested a FapD knockout in our *E. coli* secretion assay, which was unable to support secretion (Fig. 5c). A FapD construct with an OmpA signal sequence was able to restore secretion activity when expressed in the FapD knockout (Fig. 5c). This confirms that FapD is active in the periplasm. We also

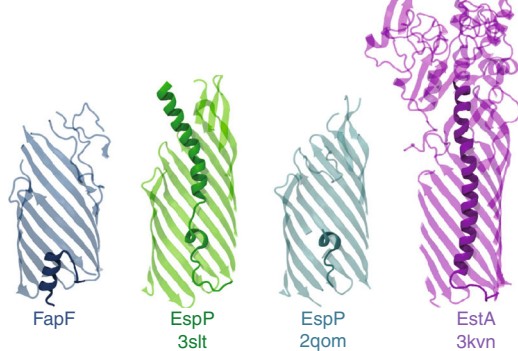

**Fig. 7** Structural comparison to the Type Va autotransporter (AT) family. The backbone root-mean-square deviation (RMSD) of FapF is below 4 Å for overlay of FapF with each of the transporter domains. EspP has been solved as the full-length pre-processed protein (3slt, *green*) and the processed, autocatalytically cleaved form (2qom, *teal*). EstA is another full-length AT with a lipase domain on the extracellular face (3kvn, *magenta*). The plug domains are shown as *cartoons* with backbones transparent and cutaway for clarity.

mutated the active site cysteine to an alanine within the operon to examine whether FapD peptidase activity is required or whether, as for the T1SS subgroup which carry aC39-like domain without catalytic activity, a substrate chaperone role is implicated[34]. Mutation of the active site cysteine in FapD (C38A) was sufficient to abolish FapC secretion, indicating that FapD is proteolytically active (Fig. 5c).

The target of FapD remains unclear and is the subject of further work. Although Fap proteins do not possess canonical C39 sequences (i.e., lv-x-GG-kag-y-ga), the presence of GG (and GA) motifs in FapE suggests that this gene product could be processed (Supplementary Fig. 18). It should be noted that as FapD is periplasmic, and this class of C39 peptidases have yet to be characterised, it may act on an alternative recognition sequence. To provide some clues to the possible targets of FapD processing, we used whole cell trypsin digestion and tandem MS peptide sequencing to determine the mature N-termini of all Fap proteins (Fig. 5d and Supplementary Data 2). These data indicated that FapE was the only protein from the operon displaying complete N-terminal processing, with no peptides observed within the first ~70 residues (Fig. 6). Intriguingly, GG and GA motifs are present immediately prior to the trypsin-sensitive start point of our FapF$_\beta$ crystal structure construct. Although MS peptide sequence coverage for the FapF transmembrane is sparse due to its membrane embedment, the coverage of the periplasmic region is restricted to peptides within the coiled-coil region and are absent from the disordered linker region, including the start point of our FapF$_\beta$ crystal structure construct. Therefore, there exists the intriguing possibility that FapD could cleave FapF and generate a fragment similar to our crystal construct that possesses a stable helical plug, thereby closing the secretion system.

## Discussion

Although the FapFs OM secretion pore is distinct from all previously described secretion systems that have been structurally characterised thus far[35], it displays individual features found in a variety of bacterial importer and exporter families. The closest related structure in the Protein Data Bank is that of the COG4313 channel, Pput2725[36], which is an importer that belongs to the meta-degradation pathway of phenol. Despite the opposite directionality of transport, the first 'PTG' motif identified in FapF is also present in Pput2725 (Supplementary Fig. 19), and

this region has been proposed to be a lateral gate that would allow the uptake of hydrophobic substrates via the membrane. While the presence of a 'PTG' motif in FapF suggest a putative lateral gate, mutagenesis of the PTG demonstrates that these are not essential for amyloid export (Fig. 4) but could provide a recognition platform for attachment.

Perhaps the most striking similarity is between the TDs of the FapF protomer and the classical Type Va autotransporter (AT) family[37], which are also characterised by a C-terminal, 12-stranded β-barrel with a helix-blocked pore in the closed state. Type Va ATs also carry a passenger domain fused to the N terminus of the β-barrel, which is released extracellularly after translocation by an autocatalytic cleavage event[38, 39]. Several structures have been solved for Type Va ATs and perhaps the most the revealing comparison can be made between FapF and the pre- and post-cleavage states of EspP (Fig. 7)[39]. The β-barrels of these EspP structures overlay with a root-mean-square deviation of 3.9 Å over 191 backbone atoms of FapF. However, in the truncated FapF$_β$ structure, the N terminus exits the barrel on the periplasmic side rather than passing completely through the pore as in ATs. Although unlikely, considering the FapF periplasmic domain is a trimeric coiled-coil, we assessed whether the N terminus would pass completely through the β-barrel, by probing the cell surface exposure of the His-tagged N terminus of full-length FapF. Western blot analysis revealed that the His-tag is accessible to antibody only after cell lysis, confirming that the N terminus lies within the periplasm (Supplementary Fig. 7). Additionally, we produced a fusion protein of FapF with an N-terminally fused β-lactamase domain between the signal sequence and coiled-coil domain. The ampicillin sensitivity of cell growth was then used to probe both correct folding of the FapF fusion and the periplasmic location of the N terminus. Expression of the fusion protein conferred ampicillin resistance indicating that a functional, N-terminal β-lactamase domain was present in the periplasm (Supplementary Fig. 7).

It is interesting to note that our single-channel current measurements show that FapF$_β$ exists in a closed state, while full-length FapF is open, indicating that the plug domain in full-length FapF is predominantly located outside of the pore. This observation is also borne out when comparing conductance measurements with those measured of the autotransporters. The conductance for the single barrel NalP was found to be 0.15 nS in 1 M KCl with the helix present within the pore, which increased to 1.8 nS when the helix was removed[40]. Our value of 4.95 nS for the trimer, i.e., 1.6 nS for a single full-length FapF subunit in 1 M KCl, is indicative of a largely open channel. It is therefore likely that the periplasmic coiled-coil and linker domains play a role in regulating pore accessibility and capturing Fap substrates. Analogous to this is the role played by the cytoplasmic, coiled-coil bundle in voltage-gated sodium channels. A conformational change within this neck domain, proximal to the pore, provide a mechanism for channel activation[41, 42]. Furthermore, coiled-coil domains have been shown to function as a platform for translocated substrates. Most notably in the sucrose OM uptake channel ScrY[43] for which it has been proposed that the trimeric coiled-coil spans the periplasm interacts with incoming maltooligosaccharides and routes them from the OM to the translocation system at the inner membrane. The FapF coiled-coil may specialise in recruiting Fap substrates to the OM β-barrel for secretion.

We tentatively propose the following mechanism of how the Fap secretion might take place (Fig. 6). FapE is found extracellularly and remains associated with the Fap fibre as a minor component[19, 44]. FapE along with FapB are essential for subsequent secretion of FapC on the bacterial surface, suggesting a role in initiating the process. MS evidence showing that the N terminus of FapE is degraded during secretion, while both FapB and FapC remain intact, may reflect the removal of a FapF secretion targeting sequence. Sequence analysis of FapE also reveals a possible domain boundary between two well-conserved N-terminal and C-terminal regions, with a disordered linker region. The C-terminal portion of FapE possesses asparagine- and glutamine-rich sequences, akin to the amyloid repeats identified in FapB and FapC (Supplementary Fig. 18). In this speculative model, FapE could therefore represent an important subunit for initiating Fap secretion. The catalytic action of the C39-like peptidase FapD, most likely on the FapE substrate, may facilitate activation of the FapF translocator complex for secretion. This model would explain observations that mutation of either FapD or FapE abolishes Fap secretion[44]. It is also plausible that FapD-promoted cleavage of FapF occurs at the transmembrane–periplasmic domain junction which also bears GG motifs (Supplementary Fig. 20) and generates the closed state observed in our crystal structure. This could represent the final state of the secretion system after subunits have been depleted from the periplasm and reinstate the OM barrier.

A prominent driving force for protein secretion across the bacterial OM (e.g., for AT passenger domains) is substrate folding on the extracellular side[45]. Furthermore, it has been shown that intrinsically disordered domains can be efficiently secreted despite the lack of folding as a possible driving force[46, 47]. The argument follows that electrostatic interactions with the periplasmic face at the base of the pore drive initial threading of the substrate into the hydrophilic lumen. The constriction subsequently allows for entropically favoured secretion by extracellular escape. This concept is highlighted in the curli functional amyloid secretion system[16], in which amyloidogenic CsgA subunits are kept unfolded and monomeric in the periplasm and then captured at the base of the CsgG pore by a capping complex with CsgE and conformationally constrained. The CsgA subunit subsequently threads and is translocated through the pore driven by an entropic expansion into the extracellular milieu and lubricated by the hydrated, charged pore lumen. Although comparison to the curli translocator CsgG shows that FapF is structurally distinct (Supplementary Fig. 3), despite secreting similar substrates, a prominent role for electrostatic interactions and spontaneous conformational changes of a disordered substrate allowing progress down an entropy-driven free energy gradient may be a shared concept[8, 47]. This is supported by the recent observation that the curli inhibitor CsgC is also a potent inhibitor of FapC amyloidogenesis and that CsgC charge mutants with diminished capacity to inhibit CsgA folding and assembly into amyloid are equally ineffective against FapC[48]. Fap substrates could be recruited by the elongated periplasmic domains of the FapF trimer (similar to CsgE-mediated capture of CsgA), which may provide the necessary conformational restriction that facilitates threading and translocation of the disordered substrates through FapF.

The *Pseudomonas* Fap biogenesis system shares key features with several classified bacterial secretions systems, namely Type I in the deployment of a C39 peptidase, Type Va in the use of a 12-stranded AT-like barrel for translocation and Curli (or Type VIII) which export similar amyloidogenic substrates. Unique features not seen in OM peptide transporters include the trimeric nature of the AT-like barrel and a prominent role in regulating secretion for a periplasmic coiled-coil. The *Pseudomonas* Fap secretion system can therefore be classed as a new hybrid secretion machinery (Fig. 6). In summary, our structural and functional characterisation of the Fap system provides new insight into the molecular mechanism underlying the secretion of disordered amyloidogenic substrates at the OM.

## Methods

**Limited proteolysis**. As described previously[21], to improve the properties of FapF for structural studies limited proteolysis was used to identify a stable fragment of the protein. Chymotrypsin, trypsin and subtilisin were used and the results indicated that stable fragments of FapF could be produced of around 35 and 31 kDa[21]. Bioinformatics predictions from DISOPRED[49] and PSIPRED[50] indicated that the N terminus has a disordered N-terminal domain of ~80 amino acids. Based on these data new FapF constructs were designed with N-terminal truncations to remove the disordered region and produce a more structured protein for study. Two of these constructs were designed based on the predicted chymotrypsin and trypsin cleavage sites corresponding to the stable fragments produced by limited proteolysis and the other was based on bioinformatics predictions of the β-strands. The corresponding region was then cloned into a series of homologues *Pseudomonas* PA7, *Pseudomonas* UK4 and *Burkholderia*, of which *Pseudomonas* UK4 produced the most promising sample.

**Cloning and expression**. *Pseudomonas* UK4 FapF$_β$ L273M and mutants, and full-length FapF were extracted and purified as described previously[21]. Briefly, the N-terminal FapF (residues 83 to 406, FapF$_β$) from *Pseudomonas* strain UK4 was cloned into a pRSF-1b vector with an OmpA leader signal sequence and an N-terminal His-tag: MKKTAIAIAVALAGFATVAQATSHHHHHHPW. This was transformed into LEMO21 cells (New England Biolabs), grown to OD$_{600}$ of 0.6–0.8 at 37 °C in autoinducing TB media before overnight induction at 25 °C. To produce a plasmid containing a FapF β-lactamase fusion (pFapF-βla) the full-length FapF construct was modified by substitution to replace the HHHHHH with the β-lactamase domain from *E. coli*. A pET46 vector containing a constitutively expressed β-lactamase was used as a positive control for ampicillin resistance. Primers used are detailed in Supplementary Data 1.

**Protein purification and crystallisation**. Protein samples were prepared as described previously[21]. Briefly, cells were harvested and resuspended in 20 mM Tris-HCl, pH 8, 1 µg ml$^{-1}$ DNase I and phenylmethane sulfonyl fluoride, followed by lysis by cell disruption (Constant Systems) at a pressure of 25 kpsi and centrifugation at 23,000×$g$ for 20 min (Ti45 rotor, Beckmann). The OM fraction was prepared by centrifugation of the supernatant at 100,000×$g$ for 2 h (Ti45 rotror; Beckmann) followed by resuspension of the pellet into 20 mM Tris-HCl pH 8 and 0.5% N-lauryl sarcosine (Thermo Fisher), stirring at room temperature for 30 min, then a second spin at 100,000×$g$ for 1.5–2 h followed by resuspension of the pellet for overnight extraction at 4 °C with 20 mM Tris-HCl pH 8, 200 mM NaCl and 1% N,N-dimethyldodecylamine N-oxide (LDAO; Sigma). FapF was then purified from the OM fraction by nickel chromatography and detergent exchanged to 0.5% C8E4 (Generon) followed by gel filtration using a Superdex-200 column (GE Healthcare). FapF$_β$ wild type and mutants for crystallography were then concentrated to 10 mg ml$^{-1}$. Protein samples used for single-channel conductance and native mass spectrometry were concentrated to ~1 mg ml$^{-1}$ before being flash frozen in liquid nitrogen and stored at −80 °C prior to use. Conditions for crystallisation were initially screened by the sitting drop method of vapour diffusion at 293 K using sparse matrix crystallisation kits (Hampton Research and Molecular Dimensions) in MRC 96-well optimisation plates (Molecular Dimensions, USA) with 100 nl protein solution and 100 nl reservoir solution using a Mosquito nanolitre high-throughput robot (TTP Labtech). Protein crystals were obtained in a reproducible manner from 100 mM sodium citrate, 20–30% (w/v) PEG400 and 100 mM NaCl. These were optimised by manual screening over sodium citrate range pH 5.5 to 6.5 in one dimension and a 50 to 100 mM NaCl concentration gradient in the second dimension using MRC 96-well plates (SwissSci Ltd) with up to 400 nl protein solution and 400 nl reservoir solution.

**X-ray data collection and processing**. Crystals were mounted in a MicroLoop (MiTeGen) and immediately flash-cooled in liquid nitrogen. The selenomethionine-substituted L273M mutant protein was used to solve the structure by selenium SAD phasing of data collected on the i03 beamline of the Diamond Light Source (DLS), UK. Diffraction data were collected with X-ray wavelength of 0.9798 Å. This single L273M mutation was introduced to increase anomalous scattering with 6 selenomethionine residues per monomer and the position chosen based on sequence alignments. This mutant was maintained for all subsequent mutations. For simplicity we refer to this mutant as the wild-type FapF$_β$ throughout this manuscript. Data were processed with XDS[51] and scaled using SCALA[52] within the Xia2 package[53]. Diffraction data of native F103A mutant were collected on the DLS i04-1 beamline at wavelength of 0.9782 Å. Data of native R157A were collected on DLS i04 beamline at 0.9795 Å. Crystallographic data are shown in Supplementary Table 1.

**Structure determination**. The data set was automatically processed and a structure solution obtained by the SHELX suite[54] using the fast_ep pipeline. All crystallographic data for FapF$_β$ are summarised in Supplementary Table 1. The content of the unit cell was analysed using the Matthews coefficient[55]. Model building was performed with Coot[56] and structural refinements were carried out with REFMAC5[57] and PHENIX[58]. The structures of F103A and R157A were solved by molecular replacement using Phaser with the L273M structure as a model[59].

Ramachandran statistics of main-chain dihedral angles (favoured/allowed/outlier (%)) are wild-type FapF$_β$ 97.1/2.5/0.5; F102A 93.5/4.6/1.8; R157A90.23/7.6/2.1.

**FapC secretion assays**. The *fap* operon from UK4 was previously cloned and ligated into PMMB190Ap for heterologous expression in *E. coli*[10]. Mutations were introduced using Q5 site-directed mutagenesis (NEB) into either the operon directly, or into the full-length FapFconstruct in pRSF-1b for complementation assays. These were verified by sequencing and then maintained in *E. coli* DH5α. Plasmids were transformed into *E. coli* BL21, and plated in overnight culture onto LB agar supplemented with 100 µg ml$^{-1}$ ampicillin for maintenance of PMMB190Ap or 100 µg ml$^{-1}$ ampicillin and 50 µg ml$^{-1}$ kanamycin for maintenance of PMMB190Ap and a complementing pRSF-1b construct. Agar was also supplemented with 1 mM isopropyl β-D-1-thiogalactopyranoside (IPTG) to induce expression of the Fap components for 48 h at 20 °C. Cells were harvested from plates and resuspended in sterile water, before recording OD$_{600}$. For each mutant, two 1 ml samples were taken and lyophilised. One of these samples was then treated with 90% formic acid and lyophilised a second time. Each freeze-dried sample was then dissolved in Tris-buffered saline buffer containing 8 M urea, using a volume normalised according to the recorded OD$_{600}$ of the original sample. Samples were then analysed by western blot using an anti-FapC antibody[60].

**N terminus accessibility assay**. Cells were grown to an OD$_{600}$ of 0.6 and induced using 1 M IPTG for 2 h at 37 °C. To produce an intact cell sample cells were diluted to OD$_{600}$ of 0.2 in 2% paraformaldehyde in phosphate-buffered saline (PBS). To produce a lysed sample cells were resuspended in 10% sodium dodecyl sulfate in PBS and boiled for 5 min. Each sample was dotted onto a nitrocellulose membrane and analysed by western blotting using anti-his horseradish peroxidase conjugate.

**Ampicillin sensitivity assay**. Cultures of *E. coli* BL21 (DE3) harbouring pFapF, pFapF-βla or pET46 with pFapF were grown to an OD$_{600}$ of 0.2 and induced overnight at 21 °C. Serial dilutions of each overnight culture were made in sterile LB media as indicated. Then, 5 µl of each dilution was spotted onto both an LB Kan and an LB Kan/Amp plate and incubated overnight at 37 °C.

**Native mass spectrometry**. The protein mixture was buffer exchanged to 0.5 M ammonium acetate pH = 7.6 containing 0.5% C8E4 utilising a biospin 6 column (Bio-Rad). Typically, 2–3 µl of the protein sample was loaded onto a gold-coated borosilicate capillary (1.2 mm outer diameter, Harvard Apparatus). Proteins were transmitted into a Q-Exactive instrument (Thermo Scientific) modified to facilitate the transmission of high mass species via a nano-electrospray ionisation apparatus. Instrument parameters were set as follows: Higher-energy collisional dissociation (HCD) pressure was maintained at $1.2 \times 10^{-9}$ mbar, capillary voltage was set to 1.6 kV and temperature to 20 °C, source and HCD energy was set to 200 V, c-trap entrance lens was set to 5.8 V, orbitrap resolution was at 17,500 and microscans were set to 10. Instrument was calibrated using cesium iodide. Data analysis was carried out using UniDec[61]. Ion-mobility mass spectrometry measurements were performed on a high mass modified Synapt high definition mass spectrometry G1 (Waters). Backing pressure was at 5.3 mBar, capillary voltage was set to 1.6 kV, cone voltage was at 200 V, trap collision energy was set to 100 V and transfer collision energy was at 20 V. Wave speed was 250 m s$^{-1}$ and data were recorded at wave heights of 9, 10 and 11 V. Data analysis was performed using Xcalibur (Thermo-Fischer) and Masslynx 4.1 (Waters) and theoretical CCS calculations were done using IMPACT[62].

**Single-channel recordings in planar lipid bilayers**. Single-channel current recordings were carried out in 0.1 M KCl, 20 mM potassium phosphate pH 7.0. A solution of 1,2-diphytanoyl-sn-glycerol-3-phosphocholine (DPhPC) (Avanti Polar Lipids, Alabaster, AL, USA) dissolved in pentane (2.5 mg ml$^{-1}$) was used to form a bilayer by the Montal–Mueller solvent-free method across a 100 µm diameter aperture in a 25 µm thick Teflon film (Goodfellow)[63]. The film separated two 0.5 ml compartments designated 'cis' and 'trans'—the cis compartment was connected to the ground and voltage, positive or negative, was applied to the trans compartment. A stock solution of FapF (0.5–1 µl of ~1 mg ml$^{-1}$) in 20 mM Tris-HCl pH 8.0, 150 mM NaCl, 1 mM dithiothreitol, 0.3% DDM was added to the cis compartment and a potential of ±200–250 was applied to induce insertion. Current flow was detected with two Ag/AgCl electrodes in 3 M KCl, 1.5% agarose bridges and amplified by a patch-clamp amplifier (Axopatch 200B, Axon Instruments, Foster City, CA, USA) connected to an Axon Instruments CV 203BU headstage. Data were filtered with a 2 kHz low-pass Bessel filter and digitised with a Digidata 1322 A converter (Axon Instruments) at a sampling frequency of 10 kHz. Data analysis was performed with pClamp 10.3 software (Molecular Devices). All measurements were done at room temperature (20.5 ± 0.5 °C).

**Molecular dynamics system setup**. Simulations were performed using the GROMACS v4.6.3 (www.gromacs.org) simulation package[64]. The protein structure converted to a coarse-grained (CG) representation using the MARTINI 2.2 force field[65]. The energy-minimised CG structure was centred in a simulation box with dimensions $100 \times 100 \times 100$ Å$^3$. A total of 250 POPE and 50 POPG lipids were

randomly placed around the protein and the system solvated and neutralised to a concentration of 0.15 M NaCl. An initial 1 μs of CG molecular dynamics simulation was applied in which a POPE/POPG lipid bilayer is spontaneously assembled around FapF$_\beta$ to give an optimised position of the protein within the membrane. The endpoint of the CG bilayer self-assembly simulation was converted back to atomic detail using a fragment-based protocol for the lipid conformations [66], while retaining the original crystal structure of the protein. Equilibration of the atomic system was achieved through 5 ns of NPT simulation with the protein coordinates restrained, before the system was subjected to 100 ns of unrestrained atomistic molecular dynamics. For the simulations with LPS a preformed membrane was used [67] and FapF incorporated using g_membed [68] by aligning the phosphate groups of the model PE/PG membranes and the assembled LPS membrane. LPS simulations used additional 0.15 M Ca$^{2+}$ ions.

**Coarse-grained simulations**. The standard MARTINI force field and its extension to proteins was used to describe all system components. During the coarse-grained self-assembly simulation an elastic network was applied to the protein using force constant of 1,000 kJ mol nm$^{-2}$ and a cutoff of 1.0 nm. Temperature was maintained at 310 K using a Berendsen thermostat with a coupling constant of τt = 1 ps, and pressure was controlled at 1 bar using a Berendsen barostat with a coupling constant of τp = 1 ps and a compressibility of $5 \times 10^{-6}$ bar$^{-1}$. Electrostatics and van der Waals interactions in the CG simulations were shifted between 0 and 1.2 nm, and 0.9 and 1.2 nm, respectively, using the standard MARTINI protocol. An integration time step of 20 fs was applied. Covalent bonds were constrained to their equilibrium values using the LINCS algorithm. All simulations were run in the presence of standard MARTINI water particles, and ions added to an approximate concentration of 0.15 M NaCl. CG LPS parameters were kindly provided by Syma Khalid (Southampton).

**Atomistic simulations**. Atomistic simulations were run using the GROMOS53a6 force field [69], and its extension to glycans [70]. The system was solvated using the SPC water model, and ions added to give a neutral system with a NaCl concentration of 0.15 M. Systems contained ~140,000 atoms including 250 POPE molecules, 50 POPG molecules, ~30,000 water molecules, 205 sodium ions and 128 chloride ions. Periodic boundary conditions were applied, with a simulation time step of 2 fs. Temperature was maintained at 310 K using a V-rescale thermostat [71] with a coupling constant of 0.1 ps, while pressure was controlled at 1 bar through coupling to a Parrinello–Rahman barostat [72] with a coupling constant of 1 ps. Particle Mesh Ewald was used for long-range electrostatics [73] and the LINCS algorithm was used to constrain covalent bond lengths [74]. Steered molecular simulations were performed as described elsewhere [75], briefly a harmonic restraint with force constant of 500 kJ was applied to the centre of mass of the 13-residue helix plug with the lipid bilayer as the reference group. A pull rate of 0.5 nm ns$^{-1}$ was used. Standard GROMACS tools were applied to analyse the simulations, with VMD [74] and PyMOL (The PyMOL Molecular Graphics System, Version 1.8 Schrödinger, LLC) used for visualisation. HOLE [76] was used to generate pore profile surfaces.

**Protein modelling**. The full-length protein was modelled by generating an idealised trimeric coiled-coil. This was done using the converged results from coiled-coil prediction software; COILS/PCOILS [23] and LOGICOIL [77] followed by CC Builder V1.0 [78]. The coiled-coil was placed at a distance from FapF$_\beta$ to result in a partially extended linker conformation and Modeller [79] was used to generate the initial conformation of the linker region using the loop_model_refine regime. This starting configuration was converted to coarse-grained format and simulated in a POPE/POPG membrane by aligning with the FapF model in order to allow the linker region to relax. The linker secondary structure was modelled as a random coil, i.e., no tertiary restraints were used in this region based on secondary structure prediction PSIPRED [80].

**Circular dichroism**. The coiled-coil peptide corresponding to residues D3 to Q40 was purchased at 97.5% quality (ChinaPeptides). Then, 1 mg was reconstituted in 0.5 ml of buffer (20 mM phosphate, 150 mM NaCl, pH 6.8) and CD measurements were carried out. Full-length FapF and FapF$_\beta$ were prepared as described for crystallography, with the final concentration of 0.5 mg ml$^{-1}$ used for CD calculations. CD was performed on a Chirascan circular dichroism spectrometer (Applied Photophysics) at 20 °C, wavelengths 200 to 260 nm, with intervals of 0.5 and 1 nm bandwidth. Samples were loaded into a quartz cuvette 100-QS with 1 mm path length. The concentration of full-length FapF was 2.4 and 45 μM for the coiled-coil peptide. The scan length per point and the number of repeats were respectively 5 s with 20 repeats for UK4 full length, 2 s with 5 repeats for UK4 coiled-coil, and 5 s with 5 repeats for the corresponding buffer of each sample. The spectra were averaged, corrected for baseline contributions, and the net spectra smoothed with a Savitsky–Golay filter (window 5) [81]. Molar ellipticities ([Θ]) are expressed in units of $10^3$ (deg cm$^2$ dmol$^{-1}$ residue$^{-1}$). Secondary-structure analyses were performed with the online server BeStSel [82]. Denaturation was monitored by CD ellipicity at wavelengths of 222 to 260 nm, and time per point was set to 2 s with 5 repeats. The temperature was increased by 1 °C per min with a tolerance of 0.2 °C starting at 20 °C and ranging to 90 °C.

**Mass spectrometry fingerprinting**. The peptide fingerprinting was performed by an in-gel tryptic digestion followed by liquid chromatography-tandem mass spectrometry (GeLC-MS/MS) approach with biological duplicates. For this, 50 μl of each bacterial culture was lyophilised, treated with 100 μl of concentrated formic acid and lyophilised again. The sample was then resuspended in 50 μl of reducing SDS–PAGE loading buffer containing 8 M of urea [19]. Insoluble material was pelleted by centrifugation for 1 min at 22,000×g and 15 μl of supernatant was loaded on a 4–20% SDS–PAGE gel (ExpressPlus, GeneScript). Electrophoresis was carried out at 50 mA for ~15 min and the gel was stained with Coomassie Brilliant Blue G250. The part of the gel that stained positive for proteins were excised into 5 fractions and subjected to in-gel digestion [83] and MS analyses, as previously described [84]. The nLC-MS setup consisted of a Ultimate3000 nano-LC (Thermo Fisher) coupled to a Q Exactive mass spectrometer (Thermo Fisher). Gradient time for each fraction was 40 min on a C18 columns (PepmapRSLC, C18, 75 μm × 50 cm, 2 μm, Thermo Fisher Scientific), resulting in 200 min for each sample. Protein identification and quantification were performed using MaxQuant [85]. Carbamidomethylation of cysteines was defined as fixed modification, and oxidation of methionines as variable modification. Digestion specificity was set to unspecific (all possible peptides of length seven and more are considered). Further settings were kept on default. This includes a peptide and protein false discovery rate below 1%. Organism-specific databases were obtained from the NCBI reference sequence database (RefSeq).

**Data availability**. The authors declare that all the data supporting the findings of this study are available within the paper and its Supplementary Information or are available from the corresponding author on request. Atomic coordinates and structure factors files have been deposited in the Protein Data Bank under accession codes (FapF TD-5o65; FapF TD F103A-5o67 and FapF TD R157A-5o68).

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

## Acknowledgements

This work was supported by the Wellcome Trust (Senior Investigator Award 100280 and multiuser equipment grant 104833 to S.M.). We thank staff at the Diamond beamline. We thank Inmaculdao Pérez-Dorado, Nessim Kichick Rodriguez, Andrea Sauerwein, Lee Sewell and Jonathon Taylor for useful discussion. We thank Firdaus Samsudin and Syma Khalid for providing LPS parameters. We thank Kurt Drickamer for use of CD machine. S.H. is supported by MRC fellowship G1100332. C.V.R. is grateful for the support of European Research Council Grant No. 695511-ENABLE.

## Author contributions

S.L.R., W.J.H., J.-L.B. and S.M. designed experiments. S.L.R., W.J.H., S.L., U.M. and W.E. designed constructs and produced protein for crystal trials. S.L.R., R.M.L.M. and S.H. determined the crystal structures. J.-L.B. performed complementation secretion assays. D.S.C. and C.V.R. performed native mass spectrometry and ion mobility experiments. S.A.I. and H.B. performed and analysed single-channel conductance recordings. S.L.R. performed molecular simulations and structure modelling. F.S. performed circular dichroism and SEC-MALS. M.D., F.-A.H., D.O. and P.H.N. performed mass spectrometry fingerprinting and analysis. S.L.R. and S.M. wrote the manuscript with input from all authors.

## Additional information

**Competing interests:** The authors declare no competing financial interests.

