## [Peer review File · Nature Communications]

Reviewer #1 (Remarks to the Author):

'A new class of hybrid secretion system is employed in Pseudomonas amyloid biogenesis' by Rouse et al

The following comments are confined to the mass spectrometry aspects of the work described in the manuscript:

The data presented in Figure 2A show clearly that the truncated protein exists as monomers, dimers and trimers, whereas the full length species exist predominantly as trimers. Figure 2 B requires more explanation: What does the relative abundance refer to? Presumably, the spectra have been deconvoluted, in which case details of the deconvolution protocol are necessary. Better still, delete this figure. The point is clear from the raw data in Fig 2A.

The ion mobility mass spectrometry data are solid and convincing.

The MS peptide sequencing is less convincing. The conclusion that FapE was the only protein displaying complete N-terminal processing was reached on the basis that no peptides were observed within the first 70 amino acid residues; however, LC MS/MS is a stochastic process and it may simply be that those peptides were not selected for fragmentation. Insufficient experimental details are given: how many replicates were performed? Which mass spectrometer platform was employed? Without that information, it is difficult to interpret the MS/MS counts represented in SFig17. Figure 5D displays sequence coverage data in a non-standard format.

Reviewer #2 (Remarks to the Author):

This is an excellent structural examination of the outer membrane protein FapF and its role in the translocation of Fap proteins across the OM by Rouse et. al. The study is focused around the X-ray crystal structure of FapF together with biophysical/functional/physiological characterization. To address the mechanism of transport a functional reconstitution of Fap amyloid translocation was established in E.coli and mutants were assessed for function.

Comments:

Questions to consider with respect to the model of FapF:

Model of transport:

In figure 7, the authors suggest that the FapF N-terminal extension is in the periplasm. What evidence is there to support the coiled-coil domain in the periplasm and not on the surface of the cell? In figure 6, the author show comparisons to ATs that have there N-termini passing through the barrel to the surface. Could a similar regulatory event occur in FapF?

In Figure 2C the authors show that full length FapF can conduct ions (probably nullifying the argument above) however the cleaved FapF(b) is plugged with no conductance. Would this represent a closed pore and one which can never be opened (Mol dynamics figure S2)? Further would the proteolysis of FapF by FapD lead to the end of secretion? Are there truncated FapF monomers found in the native outer-membrane?

How does this make sense in the middle panel of your model Figure 7 -FapF should problem not get processed prior to secretion but more likely to end secretion?

Minor comments

1. Some of the supplementary figures are not referred to in the main text.

Figure S13? S17?

2. In figure S12, mutant F126A is mentioned as an anchoring point for the plug helix?

Is this a type-O? There is no F126 -there's an F 136 and neither are mentioned in the maintext.

3. Figure 4 could use some arrows pointing to the substrate or Mw to indicate the mass.

The position of the mutants could be better displayed as a single figure indicating the position of the residues and a chart or table indicating function. The repetitive figure with no labels is uninformative. (alpha symbol in a-FapC was changed in the PDF conversion)

4. Figure 5B the overlay of FapD model and C39 are hard to distinguish -use different colors - Why would you expect these structures to be different if FapD was modelled based on C39?

“Coexpression of a Δ FapD knockout with a FapD construct with an OmpA signal sequence restores FapC secretion and confirms”

Figure 5D is hard to interpret ... What do the arrows represent?

Reviewer #3 (Remarks to the Author):

This manuscript characterizes the system involved in the secretion of amyloid in *Pseudomonas*, which is different from curli assembly in *E. coli*. The authors use a combination of biophysical and in silico methods to characterize the structure of the outer membrane protein of this system, FapF. FapF forms a trimer of 12-stranded β barrels that are gated by a short plug helix. The N-terminal portion of FapF could be modeled into a trimeric coiled coil that extends into the periplasm.

The authors convincingly show that FapF is a new prototype of transporter combining features of several other systems. Their mechanistic description however is too speculative at this stage, as many questions remain to be addressed.

Specific points

1) Fig 2C shows that the conductance measured for urea-treated FapF- β is similar to that of the full-length protein. However, the authors state in the text that single-channel recordings suggest a less constricted channel for the full-length protein (lines 162-164). What do they mean by 'less constricted'? Is it based on the small differences between the 2 proteins seen at high voltages?

2) Page 6, lines 164-168. The authors state that the trimer is important for function, but they do not show it. Disruption of the Phe-mediated contacts was made by replacing two of them by Ile residues. While the available results indicate that FapF is indeed a trimer, these replacements appear to affect the insertion of the protein in the membrane rather than its function. The authors also propose later in the text that the three pores function independently (lines 282-83).

Co-expressing wt and mutant FapF in the same cells would reveal if there are dominant negative effects indicative of the functional importance of trimerization.

3) The authors use MD simulations to address the potential function of the two PTG motifs. They observe the breaking of an H bond between strands close to the PTG motifs and propose that this might cause a gap for substrate exit or create a platform of assembly. The importance of these motifs should be addressed by other means than simulations only, i.e. mutagenesis and secretion assays.

4) The secretion assay is not presented in the proper manner. It is very difficult to compare the amounts of FapC secreted by the various strains without having immunoblots that show the amounts of FapF present in the cells in each case. This data should be presented.

For instance, the E111A mutant is said to display reduced secretion activity, which is not obvious.

One also observes differences between strains regarding the detection of FapC. For instance, FapC is absent from some mutants, and the authors speculate that these mutations place stress on the cell,

so that the FapC monomer is degraded. If this is the case, then it suggests that it is more stressful to have a non-functional FapF than no FapF at all. In a similar vein, the faint band present in the mock-treated cells and said to be intracellular FapC seems to be more intense in one of the mutants than in the wild type strain. Are these unexplained results reproducible?

Finally, the 'methods' section does not indicate that the samples were centrifuged to remove insoluble material before electrophoresis and immunoblotting. One would thus expect to see aggregated material in the stacking gel after the cells were mock-treated (no formic acid).

5) The authors mention belatedly in the text (lines 270-73) that there are two bands of secreted FapC. This should be placed earlier, when the results of the first secretion assays are presented. The explanations for the two bands are either that the protein is processed in the course of secretion, or that *E. coli* proteases generate a stable cleavage product. This should be investigated further, as the authors devote a long section of the results to the FapD protease later on. Thus, they should look at what happens in *Pseudomonas*. If immunoblotting with the FapC antibody reveals two bands in the native host, then this processing might indeed be related to the secretion process.

6) The authors show by mass fingerprinting that FapE appears to be processed at the N terminus. This is potentially interesting, but immunoblotting should be used to confirm the data. To see whether it is important for secretion, the GG site should also be mutated.

7) The authors speculate that FapF might undergo FapD-mediated proteolysis after secretion, because it has a presumed recognition site. This processing should be investigated by immunoblotting.

8) Several parts of the discussion are very speculative:

- p.11. A link between the PTG motifs and the sensing or uptake of hydrophobic quorum sensing molecules is far-fetched at this stage.

- p.13, lines 366-68. No interaction between the amyloid subunits and the trimeric coiled coil has been shown. It is thus premature to talk of an 'amyloid conveyor'.

- The barrel of FapF is said to resemble that of AT proteins. Are there other types of 12-stranded barrels that have different shear numbers and that are much more different? It seems that the comparison that the authors are trying to make between the AT mechanism and that of FapF goes too far. There is no evidence that the sequence removed from FapE is a targeting sequence to FapF, or that FapE plays a role analogous to the passenger domain of ATs. FapE is proposed to be the initiation subunit, which is possible since its deletion abolishes secretion. However, there is no information on the effect of a FapB deletion for instance. One cannot exclude other models, for instance that FapB is the initiation subunit and that the processing of FapE terminates assembly.

- Fig. 7. There is currently no evidence that the coiled coil is anchored to the peptidoglycan.

- Fig. 7. If FapF is cleaved at the end of the secretion process to close the channel, then the His tag placed at the N terminus should not be detected after amyloid assembly (cf Fig S19). Electrophoresis of cell lysates followed by immunodetection of FapF is needed to clarify this matter.

- p. 14, lines 400-403. Again, the parallel between AT secretion and amyloid assembly should not be taken too far. There is no sufficient support for the notion that the FapF structure represents an early intermediate of AT passenger secretion.

Minor points

- Please check the text and supplemental material for typos.

- F126 is mentioned in the legend to Fig. S12. Should it not be F102?

- Similarly in the legend of Fig. S13, it should be K401 instead of K400, according to Fig 1.

- Line 279. Should be F103 rather than 102.

- Lines 362-65. There is no need to refer to Fig S2 (which should be S3) as the sugar slide of ScrY is not displayed in the figure.

- Line 382: I presume there is no assembly of the substrates on the periplasmic platform of FapF prior to secretion? 'Recruitment' might be more appropriate.

Response to referees for NCOMMS-16-30126

Referee #1

We welcome the constructive comments on this work and its interest to people within the bacterial secretion field.

Reviewer #1:

The data presented in Figure 2A show clearly that the truncated protein exists as monomers, dimers and trimers, whereas the full length species exist predominantly as trimers. Figure 2 B requires more explanation: What does the relative abundance refer to? Presumably, the spectra have been deconvoluted, in which case details of the deconvolution protocol are necessary. Better still, delete this figure. The point is clear from the raw data in Fig 2A.

Response: As suggested we have deleted the Figure 2B deconvolution.

The MS peptide sequencing is less convincing. The conclusion that FapE was the only protein displaying complete N-terminal processing was reached on the basis that no peptides were observed within the first 70 amino acid residues; however, LC MS/MS is a stochastic process and it may simply be that those peptides were not selected for fragmentation. Insufficient experimental details are given: how many replicates were performed? Which mass spectrometer platform was employed? Without that information, it is difficult to interpret the MS/MS counts represented in SFig17. Figure 5D displays sequence coverage data in a non-standard format.

Response: We agree with the reviewer that the absence of identified peptides cannot be seen as concrete proof, but serves as indicative evidence for the suggested hypothesis. We clarify that the peptide fingerprinting was performed in duplicates and would like to emphasize that most of the biological cleavage sites were indicated by several peptide species. More detailed information is now provided in Figure S19. We also have added further information regarding the instrumentation in the Methods (Ultimate3000 and Q Exactive, both from Thermo Fisher) and gradient setup (40 min efficient gradient per fraction, 200 min per Sample). Figure S17 provided summed MS/MS counts from both duplicates as provided by MaxQuant. We have simplified the peptide mapping in Figure S19 and presented it in a standardized fashion. In addition we have provided additional information about the MS/MS counts for each replicate as provided by MaxQuant. We added further sequence coverage for each protein in Figure S19, which indicated very good sequence coverage (36% to 84%).

Reviewer #2:

We thank the reviewer for their support comments on this work and its interest to people within the bacterial secretion field.

General comment:

Comments:

In figure 7, the authors suggest that the FapF N-terminal extension is in the periplasm. What evidence is there to support the coiled-coil domain in the periplasm and not on the surface of the

cell? In figure 6, the author show comparisons to ATs that have their N-termini passing through the barrel to the surface. Could a similar regulatory event occur in FapF?

Response: Excellent idea based on AT similarity. We provide the supplementary Figure S7 in which we have immunoblotted for his-tag of full-length FapF in *E. coli* cells without and with cell lysis. Only lysis exposes the tag for antibody detection. We also created an N-terminal FapF fusion to a large folded domain that would not be secreted, for which we could probe for stable folding of the fusion and its presence within the periplasm. An terminal β -Lactamase-FapF fusion was created and cells tested for ampicillin sensitivity. Ampicillin resistance can be interpreted as the expression of a stable FapF fusion with an N-terminal β -lactamase domain present within the periplasm. We have modified the text and cite this supplement figure illustrating the two assays clearly. See results page 7, lines 19-25.

In Figure 2C the authors show that full length FapF can conduct ions (probably nullifying the argument above) however the cleaved FapF(b) is plugged with no conductance. Would this represent a closed pore and one which can never be opened (Mol dynamics figure S2)? Further would the proteolysis of FapF by FapD lead to the end of secretion? Are there truncated FapF monomers found in the native outer-membrane?

Response: This is a notion that we favour as well, but we are careful to not over interpret the data presented in this manuscript. Indeed MS fingerprinting of SDS-PAGE bands suggests that truncated FapF is present in native outermembrane, however these data is too preliminary and awaits the production of clean and specific antibodies. Although underway, this is beyond the scope of this manuscript. If we may, we would like to keep this comment as a suggestion

How does this make sense in the middle panel of your model Figure 7 -FapF should problem not get processed prior to secretion but more likely to end secretion?

Response: We have simplified our model in Figure 7 to improve clarity and not to over interpret our findings.

Minor comments:

1. Some of the supplementary figures are not referred to in the main text. Figure S13? S17?

Response: this showd now be corrected and figures appropriately cited.

2. In figure S12, mutant F126A is mentioned as an anchoring point for the plug helix? Is this a type-O? There is no F126 -there's an F 136 and neither are mentioned in the main text.

Response: this has now been corrected.

3. Figure 4 could use some arrows pointing to the substrate or Mw to indicate the mass. The position of the mutants could be better displayed as a single figure indicating the position of the residues and a chart or table indicating function. The repetitive figure with no labels is uninformative. (alpha symbol in a-FapC was changed in the PDF conversion)

Response: Figure 4 has be revised as suggested with arrows and markers.

4. Figure 5B the overlay of FapD model and C39 are hard to distinguish -use different colors - Why would you expect these structures to be different if FapD was modelled based on C39?

Response: As the reviewer correctly points out the model for FapD should be similar to the C39 peptidases as it was confidently homology modelled from these templates. The figure highlights better where the C39 domain is located in the cytoplasm of the ABC transporters, rather than the structural similarity.

“Coexpression of a FapD knockout with a FapD construct with an OmpA signal sequence restores FapC secretion and confirms”

Response: this sentence has been reworded in the Figure 5 caption

5. Figure 5D is hard to interpret ... What do the arrows represent?

Response: Agreed and redesigned in a more typical format

Reviewer #3:

We thank the reviewer for a thorough assessment and their comments are extremely constructive. The reviewer highlights our description of FapF as a new prototype of bacterial transporter combining features of several other systems, but would like us to back off on mechanistic speculation or provide new data. While some of the excellent experimental suggestions are out of the scope for this manuscript, we have introduced new data to help support some of our conclusions. We have also revised the discussion with this reviewer's perspective and recommendations. We believe that the new version represent a significantly improved manuscript and speculation is tempered.

Specific points:

1. Fig 2C shows that the conductance measured for urea-treated FapF- β is similar to that of the full-length protein. However, the authors state in the text that single-channel recordings suggest a less constricted channel for the full-length protein (lines 162-164). What do they mean by 'less constricted'? Is it based on the small differences between the 2 proteins seen at high voltages?

Response: Perhaps our explanation is unclear. The conductance reading should be similar however we can only measure conductance for FapF- β in the presence of urea. The interpretation for this is that the plug domain is extremely stable (as suggested from the crystal structure) and does not open sufficiently frequently to enable insertion and conductance measurements. Adding urea destabilises or partial unfolds the plug and facilitates opening. A similar observation has been made in the TonB-dependent transporters and it was postulated to mimic to some extent the in vivo plug opening. (Udho, E. et al. Proc Natl Acad Sci U S A 106, 21990-5 (2009). The text describing this has been reworded to improve clarity (Page 6, lines 10-23)

2. Page 6, lines 164-168. The authors state that the trimer is important for function, but they do not show it. Disruption of the Phe-mediated contacts was made by replacing two of them by Ile residues. While the available results indicate that FapF is indeed a trimer, these replacements appear to affect the insertion of the protein in the membrane rather than its function. The authors also propose later in the text that the three pores function independently (lines 282-83).

Co-expressing wt and mutant FapF in the same cells would reveal if there are dominant negative effects indicative of the functional importance of trimerization.

Response: Our data shows FapF is trimeric and this is mediated by two trimerisation interfaces (one from the coiled coil and one from β domain packing). We agree with the reviewer that this does not show that trimerisation is critical for secretion. We have tried dominant negative experiments with mutants on a wild-type operon as suggested and we see no effect on amyloid production, which suggests that trimerisation is either not essential or that you only need a relative few active trimeric pores for amyloid secretion and detection in our assay.

3. The authors use MD simulations to address the potential function of the two PTG motifs. They observe the breaking of an H bond between strands close to the PTG motifs and

propose that this might cause a gap for substrate exit or create a platform of assembly. The importance of these motifs should be addressed by other means than simulations only, i.e. mutagenesis and secretion assays.

Response: Agreed and we now report on secretion of a conserved PTG motif in Figure 4 and we see no effect on amyloid secretion. The text has been modified accordingly in the Discussion on page 12 first paragraph. We mention the structurally interesting aspects of the PTG motifs, but state that a direct role in amyloid secretion is not likely.

4. The secretion assay is not presented in the proper manner. It is very difficult to compare the amounts of FapC secreted by the various strains without having immunoblots that show the amounts of FapF present in the cells in each case. This data should be presented. For instance, the E111A mutant is said to display reduced secretion activity, which is not obvious. One also observes differences between strains regarding the detection of FapC. For instance, FapC is absent from some mutants, and the authors speculate that these mutations place stress on the cell, so that the FapC monomer is degraded. If this is the case, then it suggests that it is more stressful to have a non-functional FapF than no FapF at all. In a similar vein, the faint band present in the mock-treated cells and said to be intracellular FapC seems to be more intense in one of the mutants than in the wild type strain. Are these unexplained results reproducible? Finally, the 'methods' section does not indicate that the samples were centrifuged to remove insoluble material before electrophoresis and immunoblotting. One would thus expect to see aggregated material in the stacking gel after the cells were mock-treated (no formic acid).

Response: Regarding E111A being reduced we agree that this is subtle and perhaps unclear, although a difference in E111A is reproducible. To keep our argument straightforward we have removed this statement and focus on whether extracellular amyloid is present or not.

We agree that FapF expression should be tested and we have confirmed this in an optimised assay. We have experienced significant challenges in producing antibodies for the other Fap proteins that were clean and specific (this work is ongoing and beyond of the scope of this revision). We therefore redesigned our secretion assay for the mutants based around the successful complementation of the Δ FapF strain with a plasmid expressing his-tagged FapF. We recreated all our FapF mutants in this system and first demonstrated that our mutant phenotypes are recapitulated in the new mutant complementation assays (which they do). We next checked FapF expression by immunoblotting for its his-tag, which confirm that all defective mutants express FapF. See new figure 4, Figure S13 and all relevant results and discussion

The gels used here are a uniform acrylamide concentration (14%) with no stacking gel, and the gels have been cropped to remove the wells for presentation. However, as is suggested, often for the mock treated cells aggregated FapC is indeed detectable in the wells for those strains which produce amyloid, particularly when more sample is loaded. An example of a full gel with more sample loaded is given below. Properly formed amyloid which has not been treated with formic acid tends to stick to laboratory plastics, and does not necessarily form a uniform dispersion in the sample which can be easily taken up by pipette.

5. The authors mention belatedly in the text (lines 270-73) that there are two bands of secreted FapC. This should be placed earlier, when the results of the first secretion assays are presented. The explanations for the two bands are either that the protein is processed in the course of secretion, or that *E. coli* proteases generate a stable cleavage product. This should be investigated further, as the authors devote a long section of the results to the FapD protease later on. Thus, they should look at what happens in *Pseudomonas*. If immunoblotting with the FapC antibody reveals two bands in the native host, then this processing might indeed be related to the secretion process.

Response: Agreed the sentence referring to the two bands has been moved earlier to the first mention of the section assays (to page 9 lines 15 onwards). This second band is only seen when amyloid is produced, so we believe that the most likely explanation for this is an *E. coli* derived fragment of fibre FapC subunits, as in assays using host *Pseudomonas* strains the same doublet is not readily visible. We cite the earlier work in which immunoblots of FapC from *Pseudomonas* are presented and also comment in the Figure caption. We also provide Mw marker on the gels to indicate that this is a relatively small truncation.

6. The authors show by mass fingerprinting that FapE appears to be processed at the N terminus. This is potentially interesting, but immunoblotting should be used to confirm the data. To see whether it is important for secretion, the GG site should also be mutated.

Response: These are excellent suggestions but as discussed in point 4 we have yet to obtain highly specific antibodies for other Fap components (except FapC). All mutants of FapE have resulted in no amyloid secretion, so the precise role of FapE/FapE processing requires further work and is beyond the scope this study. Furthermore, as FapD is periplasmic, and this class of C39 peptidases have yet to be characterised fully, it may act on an alternative recognition sequence rather than the typical GG (or GA). Furthermore, recognition sequences usual have a hydrophobic in P4 and often a lysine in P1' which are not present in Fap sequences therefore FapD is unlikely to fit to the standard C39 class. If we may, we would like to keep ideas on FapD processing as plausible but speculative suggestions. Our discussion has been rewritten to reflect better what we do know and what appropriate speculation is. These ideas will form the basis for future experimentation by us and others in the field.

8. The authors speculate that FapF might undergo FapD-mediated proteolysis after

secretion, because it has a presumed recognition site. This processing should be investigated by immunoblotting.

Response: Again excellent suggestion. We can detect truncated FapF in native outer membranes by various MS approaches, however these data await the production of clean and specific antibodies. This work is ongoing but is beyond the scope of this manuscript. Although underway, we hope the reviewer finds it acceptable that we keep this comment.

9. - p.11. A link between the PTG motifs and the sensing or uptake of hydrophobic quorum sensing molecules is far-fetched at this stage.

Response: Agreed and the point has been removed. We have now performed mutagenesis of the conserved PTG and demonstrate that these are not essential for amyloid secretion, we focus our discussion on these being an interesting structural motif, which may play another role outside of secretion. This is mentioned in the discussion on page 12, line 10 onwards and assay shown in Figure 4

10 - p.13, lines 366-68. No interaction between the amyloid subunits and the trimeric coiled coil has been shown. It is thus premature to talk of an 'amyloid conveyor'.

Response: Agreed and a better choice of words would help to illustrate the possibility that this represent a platform for substrates. Relevant text is reworded to temper any emphasis of the point. See page 13, from line 17.

11- The barrel of FapF is said to resemble that of AT proteins. Are there other types of 12-stranded barrels that have different shear numbers and that are much more different? It seems that the comparison that the authors are trying to make between the AT mechanism and that of FapF goes too far. There is no evidence that the sequence removed from FapE is a targeting sequence to FapF, or that FapE plays a role analogous to the passenger domain of ATs. FapE is proposed to be the initiation subunit, which is possible since its deletion abolishes secretion. However, there is no information on the effect of a FapB deletion for instance. One cannot exclude other models, for instance that FapB is the initiation subunit and that the processing of FapE terminates assembly.

Response: Agreed and our AT comparison is perhaps premature. Our whole discussion of this point has been rewritten and shortened. Page 14 onwards and we present a simplified schematic Figure 7.

12- Fig. 7. There is currently no evidence that the coiled coil is anchored to the peptidoglycan.

Response: Agreed. We wanted to make the point that the linking region between the coiled coil and the barrel is long enough to span the entire periplasm, so that it is possible the coiled coil domain resides beneath the peptidoglycan layer. Reference to anchoring has been removed and this point has been reworded. Again the schematic figure has been simplified to reflect this.

13 - Fig. 7. If FapF is cleaved at the end of the secretion process to close the channel, then the His tag placed at the N terminus should not be detected after amyloid assembly (cf Fig S19). Electrophoresis of cell lysates followed by immunodetection of FapF is needed to clarify this matter.

Response: also see point 7.

14 - p. 14, lines 400-403. Again, the parallel between AT secretion and amyloid assembly should not be taken too far. There is no sufficient support for the notion that the FapF structure represents an early intermediate of AT passenger secretion.

Response: Agreed and we have removed this section on page 14.

Minor points:

- Please check the text and supplemental material for typos.

Response: checked and corrections made

- F126 is mentioned in the legend to Fig. S12. Should it not be F102?

Response: Yes – fixed

- Similarly in the legend of Fig. S13, it should be K401 instead of K400, according to Fig 1.

Response: Yes - fixed

- Line 279. Should be F103 rather than 102.

Response: Yes - fixed

- Lines 362-65. There is no need to refer to Fig S2 (which should be S3) as the sugar slide of ScrY is not displayed in the figure.

Response: reference to sugar slide has removed and the text is reworded to temper the emphasis of the point.

- Line 382: I presume there is no assembly of the substrates on the periplasmic platform of FapF prior to secretion? 'Recruitment' might be more appropriate.

Response: Agreed, recruitment is a better description and now used throughout the text.

Reviewer #1 (Remarks to the Author):

The authors have satisfactorily addressed my previous comments.